# COVID-19 Vaccine Effectiveness against Omicron Variant among Underage Subjects: The Veneto Region’s Experience

**DOI:** 10.3390/vaccines10081362

**Published:** 2022-08-20

**Authors:** Silvia Cocchio, Federico Zabeo, Giulia Tremolada, Giacomo Facchin, Giovanni Venturato, Thomas Marcon, Mario Saia, Michele Tonon, Michele Mongillo, Filippo Da Re, Francesca Russo, Vincenzo Baldo

**Affiliations:** 1Department of Cardiac Thoracic and Vascular Sciences and Public Health, University of Padua, 35131 Padua, Italy; 2Azienda Zero (Veneto Region), 35100 Padua, Italy; 3Regional Directorate of Prevention, Food Safety, Veterinary, Public Health—Veneto Region, 30123 Venice, Italy

**Keywords:** SARS-CoV-2, COVID-19, age-group of children, age-group of adolescents, VOCs, Delta, Omicron, vaccines, vaccine effectiveness, public health

## Abstract

Even if most of the complications due to COVID-19 are observed in the elderly, in Italy the impact of COVID-19 among young people has not been negligible. Furthermore, their contribution to SARS-CoV-2 circulation is still unclear. These reasons have driven policy makers to involve subjects aged 5 to 17 years in the COVID-19 vaccination campaign. However, the trade-off of vaccinating this age-group should be further investigated, especially in view of the rise of new immunologically evasive variants of concern (VOCs). We used regional databases to retrospectively estimate vaccine effectiveness over time of each approved vaccination schedule among children (5–11) and adolescents (12–17). Our findings suggest that COVID-19 vaccines were highly effective and their protection levels lasted longer during a period of Delta variant predominance, whereas they offered just mild to moderate levels of protection—apparently affected by a rapid waning effect—in a period of Omicron variant predominance. Considering these results, it is plausible to evaluate a reformulation of possible future COVID-19 vaccination campaigns among underage subjects. However, effectiveness against serious complications due to COVID-19, as well as indirect benefits of underage vaccinations, should first be addressed. Furthermore, vaccine effectiveness should be kept monitored, as new VOCs may arise, but also new adapted vaccines may start being administered.

## 1. Introduction

As of June 2022, the COVID-19 pandemic has led to more than 18 million cases, 600,000 hospitalizations, and 168,000 deaths in Italy [1,2].

Most of the COVID-19-related hospitalizations and deaths are observed in the elderly, in line with previous evidence asserting that severe complications due to COVID-19 are significantly related to age and preexisting comorbidities [3,4]. However, the impact of the disease among young people is not negligible.

In Italy, more than 4 million cases occurred in those aged under 19 years, with about 19,000 hospitalizations, approximately 400 intensive care unit (ICU) admissions, and 57 deaths due to COVID-19 registered in the same age-group [5]. In addition, the disease in children has been associated with multisystem inflammatory syndrome and neurological disorders in the scientific literature [6,7,8]. Finally, it was found that infected children—even when they rapidly recover and suffer just mild symptoms—are often affected by so-called long COVID-19 symptoms [9].

Considering the number of detected infections among subjects aged under 19, which may be significantly underestimated, especially due to the large fraction of asymptomatic or paucisymptomatic subjects among young people [10], it is likely that they significantly contribute to virus circulation in the general population. Such a thesis may be supported by evidence of a viral load in children like the one registered among adults [11] and by studies, suggesting that school reopening might have been associated with an increase in the epidemic curve [12]. However, evidence on the role of children and adolescents in boosting the epidemic provides mixed results [13,14,15,16].

Policy makers in many countries evaluated the risk–benefit ratios and progressively extended the COVID-19 vaccination campaign to underage people [17,18].

In Italy, the vaccination campaign began on 27 December 2020, and it was partially extended to underage subjects, with different timing and approved schedules, as indicated by the European (EMA, European Medicines Agency) and Italian (AIFA, Agenzia Italiana del Farmaco) regulatory agencies. Vaccines approved for underage subjects were mRNA vaccines: BNT162b2 (Comirnaty, Pfizer–BioNTech, Mainz, Germany) and mRNA-1273 (Spikevax, Moderna Biotech, Madrid, Spain). Vaccination with BNT162b2 began in March 2021 for individuals over 16 years of age [19], and then, from June 2021, it was also offered to subjects between 12 and 15 years of age with the same dosage used in adults [20]. Similarly, the mRNA-1273 vaccine was first approved for individuals over 18 years and then for subjects over 12 years by the end of July 2021, maintaining the same schedule [21]. At the beginning of December 2021, administration of the primary cycle was also extended to the 5–11 age-group, with specific formulation and dosage (i.e., BNT162b2 at one-third of the regular dose administered to adults and adolescents—10 μg vs. 30 μg) [22].

Third doses started being administered in Italy on 27 September 2021 in subjects older than 18 years using only mRNA vaccines. On 24 December 2021, BNT162b2 was approved for the 16–17 age-group and frail subjects between 12 and 15 years of age [23]. Thereafter, from January 2022, approval was extended to the whole 12–15 age-group [24].

The Veneto region is in northeastern Italy and has approximately 5 million inhabitants, of which 760,000 are underage [25]. In this area, the vaccination campaign was conducted following the national guidelines.

Despite many efforts made towards its containment, including the involvement of underage subjects in the vaccination campaign, COVID-19 still represents a threat affecting both public health and the economy of many countries, while many stakeholders are discussing further relaxation of public health measures. More evidence is needed to support the definition of strategies for containing viral spread. Particularly, sustainability of strict nonpharmaceutical interventions (NPI), especially those involving child- and adolescent-oriented services (such as school closures), have been widely debated, considering its psychological and social impact [26,27].

Furthermore, assessment of the present epidemiological situation must focus on the currently predominant variants of concern (VOCs), since there is increasing consensus that new VOCs are generally characterized by milder clinical manifestations and increased transmissibility [28,29]. The latter may be related to mechanisms of immunological evasion leading to a fall in the protection afforded by COVID-19 vaccinations and previous infection [30,31,32].

In this context, the trade-off of vaccinating young people is less clear, and determining effectiveness of COVID-19 vaccinations specifically in this subpopulation is crucial to gain more insights on costs and benefits for evaluating and possibly reshaping public health policies [33].

The aim of the present study was to investigate the vaccination’s impact on underage subjects in the Veneto Region during a period characterized by a prevalence of the Omicron (B.1.1.529) variant (and sublineages), which (since January 2022) is the predominant one in Italy [34]. To give a broader picture on how vaccinations impacted as a whole on this specific subpopulation, we also estimated the effectiveness of a primary cycle among adolescents during the predominance of Delta (B.1.617.2). Particularly, we provide estimates for vaccine effectiveness stratified per vaccination status and timing from the last vaccination among two different age-groups: 5–11 years and 12–17 years.

## 2. Materials and Methods

We conducted the same retrospective observational analysis in two distinct periods: from 1 August 2021 to 25 October 2021 (which we will henceforth refer to as the Delta period), and from 1 February 2022 to 27 April 2022 (which we will refer to as the Omicron period). These two periods were chosen in order to represent the homonymous most prevalent VOC, with respect to the main steps of the COVID-19 vaccination campaign in children and adolescents. We assumed representativity by proxy of sampled swabs sequenced according to the regional viral genome monitoring project [35]. Actually, from August to October 2021, more than 99.9% of sequenced samples in the Veneto region belonged to the Delta variant [36,37] whereas from February to April 2022, more than 98.8% of sequenced samples on the Veneto region belonged to the Omicron variant and sublineages [38,39].

For both analyses, anonymized data were extracted from regional databases containing general information about all the RT-PCR and rapid antigen tests for SARS-CoV-2 and the COVID-19 vaccinations performed by the Veneto region, as well as the monitoring of confirmed SARS-CoV-2 cases taken under charge of the same region.

Analyses were focused on underage inhabitants of the Veneto region eligible to receive COVID-19 vaccinations, which coincided with the 5–17 age-group. From now on, we define children as those aged 5–11 and adolescents those aged 12–17. These age-groups have been chosen in order to reflect the underage categories identified by nationally approved vaccination schedules.

First, we took the census of the Veneto region population by age-group from the Italian National Institute of Statistics (ISTAT) to estimate the vaccine coverage and the share of subpopulations under study that were tested during the two periods. We used census data updated to 2021 for the Delta period and 2022 for the Omicron period. Thus, we used vaccination and swab databases, respectively, to count the number of subjects belonging to a specific age-group who were vaccinated before the period’s baseline and to compute how many of them were actually tested during the considered period.

To compute the incidence rate (IR) of infection according to vaccine status and to estimate the vaccine effectiveness (VE), we restricted our analysis to those subjects aged between 5 and 17 years who were tested at least once during the two periods considered in the study. Hence, we deterministically linked the databases through an anonymous alphanumeric code that uniquely identifies each subject. Since rapid antigen tests had the same diagnostic value as RT-PCR in the Veneto region from the end of December 2021 [40], we defined “infection” as any first positive test, either RT-PCR or rapid antigen. We defined “reinfection” as any positive test occurring after at least 90 days from a preceding infection. Positive tests performed within 90 days from an infection were all counted as a single episode. We arbitrarily decided to keep the same criteria for defining infection during the Delta period, even though national recommendations stopped requiring RT-PCR after a positive rapid antigen test as a confirmation test for diagnosis in December 2021. Subjects who turned 5 or 18 years old during the considered time intervals were respectively belatedly included or prematurely censored (right censoring) from the study, in order to keep into account their presence just for the number of days in which their age matched the study design.

We did not include in the study population those tested positive within 90 days before they entered the study. Furthermore, the following conditions were needed for a subject to be included in the study population, to avoid considering vaccination schedulesnoncompliant with the standards currently approved by the EMA: receiving the first dose after his/her fifth birthday, receiving the third dose after his/her twelfth birthday, receiving BNT162b2 vaccines for subjects between 5 and 11 years old and BNT162b2 or mRNA-1273 vaccines for those between 12 and 17, receiving the second dose at least 21 (for BNT162b2) or 28 (for mRNA-1273) days after the first one, receiving the third dose with BNT162b2 after at least 120 days from the second one, and receiving a homologous primary cycle.

We provide estimates for the effectiveness of primary cycle (two-dose) and booster-dose regimen. Moreover, for the Delta period, we provide an estimate for effectiveness of just the primary cycle among adolescents, because in that period it was the only approved vaccination schedule among underage subjects between 12 and 17 years.

The descriptive characteristics of such study populations at the baseline were summarized through percentage, medians with interquartile ranges (IQRs), and means with 95% confidence intervals (CIs).

From study entry, subjects belonging to the cohort populations were followed up over time. Changes in the time-dependent covariates, such as age-group, vaccine status, and time from vaccination were recorded as well as SARS-CoV-2 infections, right censoring, and number of days of follow-up (FU).

Thus, an extension of the proportional hazard Cox regression model capable of dealing with time-dependent covariates [41] was run considering SARS-CoV-2 infection as the outcome, sex, vaccine status and presence of a prior infection as independent covariates, and age-group, vaccine typology, and time from the last dose received as strata. As previously proposed [42], the time from the last dose was categorized into intervals of 0–6 days, 7–13 days, 14–34 days, 35–69 days, and 70+ days.

For each combination of strata, the corresponding IR per 10,000 person-days and VE are presented. IR was computed as the number of cases in a specific stratum over the number of person-days of FU in the same group, whereas VE was estimated with the formula
*VE* = (1 − *HR*)%(1)
where *HR* is the adjusted hazard ratio estimated by the model for the risk of contracting a SARS-CoV-2 infection between vaccinated (with either primary cycle or booster-dose regimen) and unvaccinated subjects belonging to the same stratum’s age-group. Overall incidence and effectiveness irrespective of either type of vaccine administered or the timing of the last dose received is provided by grouping subjects according just to the number of doses received.

Confidence intervals with a significance level equal to 5% (95% CI) were computed for each *VE*’s estimate. When an estimate turned out not to be statistically significant, i.e., the Wald test accepted the null hypothesis that the corresponding unexponentiated coefficient was not statistically different from 0 and consequently the *VE* confidence interval crossed 0, we indicate it as just nonsignificant (ns). Statistical significance levels in differences of *VE* across different vaccine statuses were inferred through the nonoverlapping of the corresponding confidence intervals. Both data manipulation and statistical analysis were performed with specific libraries of Python (Python 3.9.7), which allowed us to deal with a large amount of data with a low execution time. Figure 1 and Figure 2 were generated with Microsoft PowerPoint and Microsoft Excel, respectively.

## 3. Results

### 3.1. Vaccine Coverage and Characteristics of Study Populations at the Baseline

In 2021 the 5- to 17-year-old Veneto population was composed of 585,546 subjects: 303,045 children and 282,501 adolescents. As of August 1, 2021, there were no vaccinated subjects in the 5–11 age-group, whereas 27.1% (N. 76,432) of the 12–17 age-group had already received their first COVID-19 vaccination. Approximately 24% (N = 73,390) of the children were tested during the Delta period, whereas 51.3% (N = 144,843) of adolescents received a test during the same time interval.

After applying eligibility criteria, the number of subjects enrolled in the study population as of the Delta period’s baseline amounted to 212,299. The median number and the mean value of swabs for each subject belonging to the study cohort were respectively equal to 1 (IQR 1–2) and 1.91 (95% CI 1.90–1.91), with an overall of 404,702 swabs performed among the whole population. Among all these swabs, 77.6% (N = 314,123) were rapid antigen tests, whereas the remaining were RT-PCR ones.

As of 25 October 2021, which coincides with the end of Delta period, 3.6% of the study population (N = 7637) subjects had tested positive. Just 1.7% (N = 133) of all the detected infections occurred among subjects who had already been positive before, regardless of vaccination status.

In 2022 the 5–17 age-group was composed of 578,609 subjects, with 295,455 children and 283,153 adolescents. As of 2 February 2022, approximately 31% (N = 92,989) of the 5–11 age-group had received the first COVID-19 dose, whereas 79.8% (N = 225,983) of the 12–17 age-group was at least partially immunized. Approximately 53.5% (N = 158,171) of the children were tested during the Omicron period, whereas 42% (N = 11,892) of adolescents received a test during the same time interval.

After applying eligibility criteria, the number of subjects enrolled in the study population as of the Omicron period’s baseline amounted to 218,285. The median number and the mean value of swabs for each subject belonging to the study cohort were respectively equal to 2 (IQR 1–2) and 1.91 (95% CI 1.91–1.92), with an overall of 417,466 swabs performed among the whole population. Among all these swabs, 95% (N = 396,526) were rapid antigen tests and the remaining RT-PCR ones.

As of 27 April 2022, which coincides with the end of the Omicron period, 34.2% of the study population (N = 74,699) had tested positive. Just 5.4% (N = 4005) of all the detected infections occurred among subjects who had already been positive before, regardless of vaccination status

Descriptive characteristics of such study population are summarized in Table 1.

### 3.2. Incidence of SARS-CoV-2 Infections per Vaccine Status and Vaccine Effectiveness

A flowchart summarizing the incidence rate of SARS-CoV-2 cases in both the considered periods stratified by age-group and vaccine status is presented in Figure 1.

The overall IR was 4.3 versus 4.9 cases per 10,000 person-days during the Delta and Omicron periods, respectively.

During the Delta period, there were no vaccinations among children, and 3835 cases were recorded in this subgroup, resulting in an IR equal to 6.5 cases per 10,000 person-days. In the same period, 3802 cases were recorded among adolescents (3.2 cases per 10,000 person-days), with 81% (N. 3078) of them occurring among unvaccinated individuals. Unvaccinated adolescents turn out to have an IR of 4.3 cases per 10,000 person-days, significantly higher than that registered among partially (2.9 cases per 10,000 person-days) and fully (0.8 cases per 10,000 person-days) immunized adolescents.

During the Omicron period, 41,288 and 33,413 infections were detected among children (49.4 cases per 10,000 person-days) and adolescents (50.7 cases per 10,000 person-days), respectively. The IR of unvaccinated children was 51.6 cases per 10,000 person-days, which turns out to be significantly higher than the one registered among fully immunized subjects belonging to the same age-group (44.2 cases per 10,000 person-days). Among adolescents, IR of unvaccinated subjects was 55.1 cases per 10,000 person-days, which is slightly lower than that among those fully immunized (56.1 cases per 10,000 person-days), but significantly higher than the IR recorded among adolescents who had received their third dose (43.9 cases per 10,000 person-days).

The overall effectiveness in both the Delta and Omicron periods (irrespective of the type of vaccine) of any combination of vaccine status and timing of the last dose received is presented in Figure 2. It is worth mentioning that colors in Figure 1 just highlight the categories for which overall effectiveness is reported in Figure 2.

During the Delta period, adolescents who had completed their primary cycle had an overall protection of 85% (79–90), which is substantially stable with respect to timing from the last dose.

During the Omicron period, the estimated effectiveness of a primary cycle significantly decreased in time, resulting in an overall effectiveness equal to 35% (34–37) among children and 19% (16–21) among adolescents. Third doses, available just for adolescents during the Omicron period, provided an overall VE of 38% (36–40). The VE of a third dose regimen was above 72% (70–73) within 35 days from the vaccination, whereas after more than 70 days their protection was not significant anymore.

The IR and corresponding VE among adolescents during both the Delta and the Omicron scenarios and for each combination of vaccination status, type of vaccination received, and number of days elapsed from the last dose received are presented in Table 2. 

During the Delta period, the overall effectiveness afforded by an mRNA-1273 primary cycle was 94% (88–97) whereas by a BNT162b2 primary cycle it was 86% (84–88). The difference in the afforded protection was not statistically significant. Both the vaccines provided generally high protection, with effectiveness above 85% (83–87), and just minor downward fluctuations occurred when the last dose had been received more than 70 days before the observation.

In the Omicron scenario, the overall effectiveness afforded by a primary cycle with BNT162b2 among children was 35% (34–37) and it ranged from 72% (69–74) in the first week from vaccination to 23% (20–26) after more than 70 days from vaccination. The primary cycle with the same vaccine among adolescents provided an overall VE of 17% (15–20) with an initial value of 81% (76–85), which decreases until 8% (5–11). The overall protection afforded by a primary cycle with mRNA-1273 among adolescents was 30% (26–33), ranging from 88% (81–92) in the first week to 20% (15–24) after more than 10 weeks. A third dose regimen with BNT162b2, available just for adolescents, provided an overall VE of 38% (36–40), even if the estimated protection was no more significant after more than 70 days from the vaccination. A heterologous third dose regimen with BNT162b2 after a primary cycle with mRNA-1273 provided an overall VE of 53% (50–56) with a protection level of 82% in the first week, which turned out to be no more significant after more than 10 weeks.

## 4. Discussion

We used permanent census data and regional databases of COVID-19 vaccinations to estimate vaccine uptake among children and adolescents in Veneto in both the Delta and Omicron periods. As of 1 August 2021, we observed a vaccine coverage of 27.1% in adolescents aged 12–17. As of 1 February 2022, the vaccine coverage among the same age-group increased to 79.8%. From December 2021, children aged 5–11 were eligible for vaccination and by 1 February 2022, 31% of them had already received their first dose. However, vaccine coverage in this latter age-group did not increase substantially over the following months. Such numbers show a discrete adherence to the vaccination program among adolescents, whose vaccine coverage is slightly lower than among the general Veneto region population [43]. On the contrary, our results show a scarce vaccine uptake among children, which is possibly because they have been just recently involved in the vaccination campaign, even if several studies have also highlighted the presence of parental hesitance [44,45,46].

According to our findings, during the Delta period the incidence of infections among adolescents was inversely proportional to the number of doses received. Particularly, we found an IR among partially vaccinated subjects and those who completed their primary vaccination cycle about 1.5 and 5 times lower than that among unvaccinated adolescents, respectively (2.9 and 0.8 vs. 4.3 cases per 10,000 person-days). Interestingly, the children’s IR was the highest registered in the Delta period—when only adolescents were eligible to receive COVID-19 vaccinations—with 6.5 cases per 10,000 person-days. This number was higher than the 4.3 cases per 10,000 person-days observed in unvaccinated adolescents within the same time period. It is reasonable to assume that the latter were less threatened by SARS-CoV-2 infections than children, due to the indirect benefits of COVID-19 vaccinations administered to their peers and schoolmates. Indeed, the IR observed before vaccination campaigns started being rolled out appeared to be generally higher in adolescents [47].

During the Omicron period, the IR among unvaccinated children was 51.6 cases per 10,000 person-days, significantly higher than that registered among fully immunized subjects belonging to the same age-group (44.2 cases per 10,000 person-days). No decrease in the incidence of infections was found among adolescents who had had two doses of vaccine when compared with unvaccinated subjects (56.4 vs. 55.1 cases per 10,000 person-days, respectively). This information should be carefully considered, keeping in mind that many adolescents had received their primary cycle months before the beginning of the Omicron period. For instance, the higher IR and the relatively lower VE that we observed in adolescents previously vaccinated with BTN162b2 when compared with other vaccine schedules is likely related to a longer time elapsed from the vaccination for a significant proportion of individuals. Actually, IR registered among adolescents who had received their third dose was significantly lower than that among unvaccinated ones (43.9 vs. 55.1 cases per 10,000 person-days). Such findings may suggest that in the Omicron scenario just subjects who had completed the most recently approved vaccination schedule (primary cycle for children, three-dose regimen for adolescents) were likely to be slightly less threatened by SARS-CoV-2 infections than unvaccinated subjects belonging to the same age-group.

Our results on vaccine effectiveness against infection among underage subjects support and enrich previous findings in showing that COVID-19 vaccines were highly effective in the Delta scenario [48] whereas they provided just mild to moderate protection against Omicron infection [49,50,51]. During the latter period, the estimated overall effectiveness of a primary cycle among children was significantly higher than among adolescents. However, effectiveness comparisons adjusted for timing from the last dose did not lead to the same conclusion. Conversely, in the first 2 weeks from the second dose adolescents were significantly more protected, then VE decreased sharply over the course of the following weeks. Eventually, when more than 70 days had elapsed from the vaccination, VE appeared to be lower in adolescents than in children. It is difficult to assert that the observed differences in the first 2 weeks should be attributed to the different dosage—since this trend vanished from the third week—whereas it is reasonable to argue that the overall discrepancy may be mainly due to the fact that adolescents were included in vaccination campaigns much earlier than children, as mentioned before.

Our analysis of the VE trend during the Delta period enriches the evidence that protection levels among adolescents remained quite stable over time, with just minor fluctuations occurring after more than 70 days from vaccination [52]. On the contrary, in the Omicron period, for both the age-groups, the effectiveness of a primary cycle was initially medium–high, but rapidly waned with low levels of protection occurring when more than 35 days hadelapsed from vaccination. In light of the quasi-constant trend for VE during the Delta period, it is very possible that the waning effect in the Omicron period was rather due to mechanisms of immunological evasion of Omicron and its subvariants than due to a natural contraction over time of the immunity.

In agreement with previous scientific evidence, our results confirm that BNT162b2 and mRNA-1273 booster doses are effective in restoring afforded protection to levels similar to those provided by the primary cycle in children and adolescents during the period of Omicron predominance [51]. Yet, protection afforded does not seem to last over time, as reported in the literature [53,54], and this behavior occurs regardless of age, with effectiveness after 5 weeks dropping to mild levels, as happened for primary cycles. This finding may unfortunately suggest that currently administered vaccines cannot provide a long-lasting and satisfactory solution in reducing the spreading of the new Omicron (and sublineages) VOC.

When further stratifying analysis in adolescents by the brand of administered vaccines, we did not find clear-cut differences in protection levels afforded by different primary cycle schedules. In fact, many differences were negligible or not significant and others cannot lead to robust assertions, since again they were not confirmed when comparing VE under the same time interval after the administration of the last dose. Thus, it is reasonable to conclude that there are no significant differences in effectiveness of BNT162b2 and mRNA-1273 vaccines.

We believe that our study has several important strengths and added values. To begin with, we conducted statistically significant and robust analysis on large volumes of regional data. The generality and amplitude of information on which the study is based make our analysis easily reproducible and scalable and minimize the probability of potential selection bias. Secondly, according to our knowledge, the present study is one of the first (both in Italy and globally) providing vaccine-effectiveness estimates against Omicron and Delta infections among all underage subjects eligible to receive the COVID-19 vaccination. Considering both these latter subpopulations allowed us to do a comparison between two subgroups that receive different formulations and dosage of vaccines and to provide insights—which may hopefully be useful to policy makers—on the whole population attending school currently involved in the COVID-19 vaccination campaign.

It is also worth mentioning that we presented overall as well as stratified per vaccine brand and timing from last-dose-received effectiveness, making it possible to compare the protection afforded by different vaccination schedules and investigate the temporal trend in the effectiveness of COVID-19 vaccines.

Nonetheless, the study is not without limitations that deserve to be mentioned, including the retrospective methodology. The main limitation is that vaccine effectiveness estimates may be biased by several immeasurable factors. Firstly, even considering both rapid antigen and RT-PCR tests, it is plausible that many cases were underestimated, for at least two reasons: a relatively lower sensitivity of rapid antigen tests [55,56] and a greater number of asymptomatic individuals within the age-groups considered. In addition, specific information, such as ethnicity and preexisting conditions, was not available, meaning that we may have not considered all potential confounding factors in our estimates. Finally, information about possible deaths not due to COVID-19, though rare in our study populations, or relocations outside the Veneto region were not at our disposal, meaning that we may have not censored subjects who prematurely exited the study.

The overlapping confidence-interval methods by which we assessed significance of differences among protection afforded by different vaccine status was naïve and conservative, meaning that—although unlikely—in some comparisons we may have missed significant differences between two estimates with overlapping CIs.

It is also important to keep in mind that our findings on the comparison of VE during the Delta and Omicron periods have to be interpreted cautiously, since estimates are results of two independent analyses performed in periods characterized by different epidemiological scenarios and non-identical public health policies, including those regarding the diagnostic value of rapid antigen tests. Also, comparisons between children and adolescents should be made with caution, as social behavior and COVID-19 regulations might impact the IR differently between these two groups. Finally, it is worth mentioning that the vaccine status of subjects in our study population changed over time, and to keep this into account we borrowed methods from survival analysis while we neither tested differences nor performed regressions with baseline characteristics, meaning that Table 1 merely provides a description of the study populations as of the baseline, and it does not give any other information on the risk of being infected.

All that said, we think that our findings should be further investigated and deepened, in order to gain an exhaustive understanding of how vaccinations among underage subjects comprehensively reduce the SARS-CoV-2 burden and to provide enough elements to possibly evaluate a reformulation of future eventual COVID-19 vaccination campaigns.

For instance, we believe that indirect benefits of involving underage subjects in the COVID-19 vaccination campaign should be investigated. Particularly, the role of COVID-19 vaccines among children and adolescents in preventing or limiting the spreading in school and family clusters and its potential consequences should be clarified.

Also, the effectiveness of vaccines in preventing serious COVID-19-related complications, especially among frail children and adolescents, should be considered. Regarding this aspect, it is important to note that we were not able to assess this doubt, since among all positive subjects found during the study, none needed to be hospitalized and none died.

Finally, we think that vaccine-effectiveness estimates should be continuously monitored, since the waning effect of vaccines already administered may become even more evident and new VOCs may arise, as well as new vaccine formulations being updated to be immunologically adapted to the predominant VOC.

## 5. Conclusions

In conclusion, we have shown that COVID-19 vaccines significantly protected adolescents from Delta infections, whereas in the Omicron scenario their effectiveness among children and adolescents was affected by a rapid waning effect—probably sped up by the continuous rise in new Omicron subvariants—and became mild to moderate 35 days after vaccination. Our findings suggest that third doses restore the protection levels just temporarily, while we did not find any remarkable evidence indicating that the reduced dosage administered to children significantly affects VE. Such findings make it plausible to consider a reformulation of potential future COVID-19 vaccination campaigns among underage subjects, even if indirect benefits provided by involving children and adolescents in the vaccination campaign and vaccine effectiveness against serious complications due to COVID-19, especially among frail underage subjects, should be deepened. Furthermore, VE should be continuously monitored to account for the possible rise of new VOCs and the development of new vaccines immunologically adapted to the predominant VOC.

## Figures and Tables

**Figure 1 vaccines-10-01362-f001:**
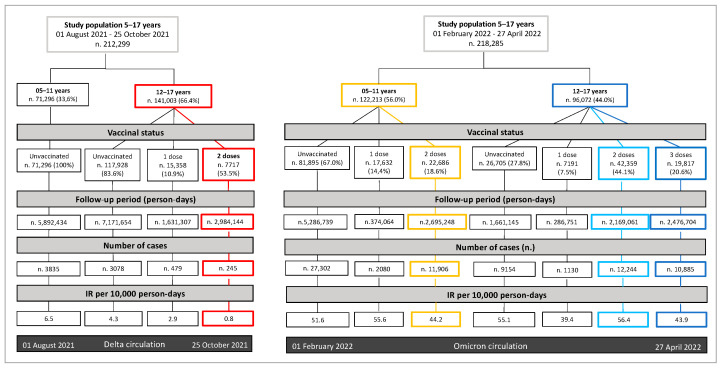
Flowchart comparing incidence proportion per vaccine status in both Delta and Omicron periods.

**Figure 2 vaccines-10-01362-f002:**
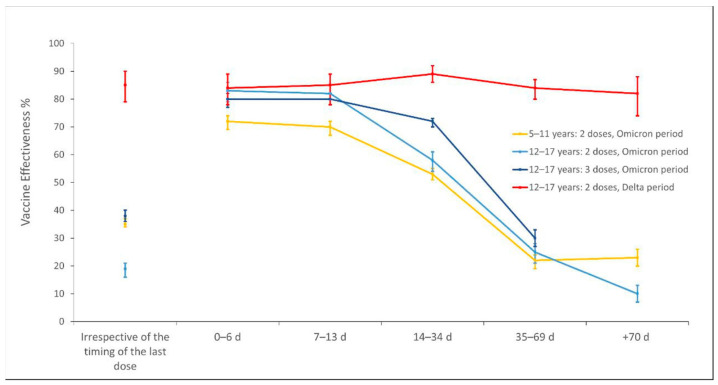
Overall effectiveness of vaccination status stratified by the timing of the last dose received.

**Table 1 vaccines-10-01362-t001:** Characteristics of the baseline study population: overall, negatives, and positives (tested positive during the study population). In the first column, covariates’ names are indicated in bold (*), covariates values (macrogroups) are presented in italic (**), whereas further subgroups of covariates value are formatted in standard text (***).

Characteristic of the Baseline Study Population	Delta Period 1 August 2021–25 October 2021			Omicron Period 1 February 2022–27 April 2022		
	Overall (N = 212,299)	Negatives (N = 204,662)	Positives (N = 7637)	Overall (N = 218,285)	Negatives (N = 143,586)	Positives (N = 74,699)
**Percentage over the whole population**	100%	96.4%	3.6%	100%	65.8%	34.2%
**Sex** *						
*Females* **	98,277	94,734 (96.4%)	3543 (3.6%)	106,038	68,623 (64.7%)	37,415 (35.3%)
*Males*	114,022	109,928 (96.4%)	4094 (3.6%)	112,247	74,963 (66.8%)	37,284 (33.2%)
**Age class**						
*05–11*	71,296	67,461 (94.6%)	3835 (5.4%)	122,213	80,927 (66.2%)	41,286 (33.8%)
*12–17*	141,003	137,201 (97.3%)	3802 (2.7%)	96,072	62,659 (65.2%)	33,413 (34.8%)
**Prior infection**						
*No*	199,435	191,391 (96.2%)	7504 (3.8%)	198,871	128,177 (64.5%)	70,694 (35.5%)
*Yes*	12,864	12,731 (99%)	133 (1%)	19,414	15,409 (79.4%)	4005 (20.6%)
**Vaccine status**						
*Unvaccinated*	189,223	181,876 (96.1%)	7347 (3.9%)	108,6	71,124 (65.5%)	37,476 (34.5%)
*First dose*						
BNT162b2 ***	15,181	14,953 (98.5%)	228 (1.5%)	22,777	16,439 (72.2%)	6338 (27.8%)
mRNA-1273	178	175 (98.3%)	3 (1.7%)	2046	1490 (72.8%)	556 (27.2%)
*Second dose*						
BNT162b2	7671	7612 (99.2%)	59 (0.8%)	56,83	37,086 (65.3%)	19,744 (34.7%)
mRNA-1273	46	46 (100%)	0 (0%)	8215	5408 (65.8%)	2807 (34.2%)
*Third dose*						
BNT162b2				18,353	11,170 (60.9%)	7183 (39.1%)
Primary cycle with mRNA-1273 + BNT162b2				1464	869 (59.4%)	595 (40.6%)

**Table 2 vaccines-10-01362-t002:** Incidence (per 10,000 person-days) and adjusted vaccine effectiveness stratified per age-group, vaccine status, and number of days from the last dose received. In the first column, age groups among which VE is estimated are presented in bold (*), vaccinal status is presented in bold+italic (**), the specific vaccination schedule is reported in italic (***) and the corresponding timing from the last dose received is formatted in standard text (****).

Vaccine Status and Number of Days from the Last Dose Received, Stratified by Age Class	Delta Period		Omicron Period	
	Incidence (×10,000 Person-Days)	Vaccine Effectiveness % (95% CI)	Incidence (×10,000 Person-Days)	Vaccine Effectiveness % (95% CI)
**Age class 05–11** *				
*Unvaccinated* **			51.6	Ref.
*Second dose*				
*BNT162b2* ***				
Irrespective of the timing of the last dose ****			44.2	35 (34–37)
0–6 d			33.4	72 (69–74)
7–13 d			35.6	70 (67–72)
14–34 d			36.6	53 (51–55)
35–69 d			46.8	22 (19–24)
+70 d			56.9	23 (20–26)
**Age class 12–17**				
*Unvaccinated*	4.3	Ref.	55.1	Ref.
*Second dose*				
*Any vaccine irrespective of the timing of the last dose*	0.8	86 (84–88)	56.4	19 (16–21)
*BNT162b2*				
Irrespective of the timing of the last dose	0.9	85 (83–87)	57.5	17 (15–20)
0–6 d	1.0	83 (76–88)	24.5	81 (76–85)
7–13 d	1.0	84 (77–89)	21.7	83 (79–86)
14–34 d	0.7	88 (85–91)	34.9	59 (55–62)
35–69 d	1.0	83 (79–87)	51.2	23 (19–27)
+70 d	1.0	82 (74–88)	65.1	8 (5–11)
*mRNA-1273*				
Irrespective of the timing of the last dose	0.4	94 (88–97)	52.8	30 (26–33)
0–6 d	0.6	90 (69–97)	15.9	88 (81–92)
7–13 d	0.6	90 (68–97)	29.4	78 (69–84)
14–34 d	0.3	96 (86–99)	40.3	55 (49–61)
35–69 d	0	ns	49.6	29 (23–35)
+70 d	0	ns	58.5	20 (15–24)
*Third dose*				
*Any vaccine irrespective of the timing of the last dose*			43.9	38 (36–40)
*BNT162b2*				
Irrespective of the timing of the last dose			44.3	38 (36–40)
0–6 d			27.8	79 (77–81)
7–13 d			26.0	80 (78–82)
14–34 d			24.7	72 (70–73)
35–69 d			48.4	30 (27–33)
+70 d			77.0	ns
*Primary cycle with mRNA-1273 + BNT162b2*				
Irrespective of the timing of the last dose			40.4	53 (50–56)
0–6 d			23.9	82 (77–86)
7–13 d			26.4	80 (75–84)
14–34 d			25.5	71 (66–74)
35–69 d			48.1	32 (26–38)
+70 d			79.9	ns

## Data Availability

The data supporting the findings of this study are available from the corresponding author upon reasonable request, and first has to be approved by Azienda Zero (Veneto region).

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
