# Peer review of "COVID-19 Vaccine Effectiveness against Omicron Variant among Underage Subjects: The Veneto Region’s Experience"

_vaccines, 2022, doi:10.3390/vaccines10081362_

Round 1

Reviewer 1 Report

The subject of the publication is very current and exciting. It is excellent that the authors investigated the trade-off of vaccinating this age group, especially in view of the rise of new immunologically evasive variants of concern (VOCs).

But the paper contains significant gaps and ambiguities.

Reviewer's comments:

In my opinion, the keywords should be modified - add "age groups of children" instead “children” - attention to the negotiation!

“Policymakers in many countries” – (examples and references from other countries should be given, e.g., Germany, Poland, France) – “evaluated the risk-benefit ratios and progressively extended the COVID-19 vaccination campaign to underage people” – (it is worth citing other papers except 17-20 - do the papers concern only Italy?). 

“In this context, the trade-off of vaccinating young people is less clear, and determining the effectiveness of COVID-19 vaccinations specifically on this subpopulation is crucial, to gain more insights on costs and benefits for evaluating and possibly reshaping public health policies”. – cite relevant items in pharmacoeconomics (e.g., cost-effectiveness, or EMA or HTA guidelines for public health policies). 

“The descriptive characteristics of such study populations at the baseline were summarized through percentage and median with their respective interquartile range (IQR).” – it is also worth adding the value of means (the median does not always give a complete description of the data).

Materials and Methods

“Confidence intervals with a significance level equal to 5% (95% C.I.) were computed for each VE’s estimate. When an estimate turned out to be non-statistically significant, we indicated it just as non-significant (ns)”. – no information on the type and scope of the statistical tests used and the names of the software (except for the specific libraries of Python (Python 3.9.7). If only Python 3.9.7 is used, full justification should be given why. 

For the percentage results, you can use the structure tests - table 1 - a complete lack of statistical comparisons - they can be very interesting.

Throughout the study, the levels of statistical significance should be given (sometimes even with no statistical difference) - then you can indicate the so-called statistical trends.

Figure 1 - no information about the individual colors' meaning (red, yellow, blue, navy blue).

Figures 1 and 2 - no information about the statistical significance or no statistical significance of it (the graph suggests that such significance exists). 

“We did not find clear-cut differences in protection levels” - “clear-cut differences” - question: are there any statistical differences? You have to write: clear-cut differences or not clear-cut differences or no statistical difference? Statistics is a very clear science.

“We conducted statistically significant and robust analysis on large volumes of regional data.” - please indicate exactly which data is a statistically significant difference.

Conclusion: we showed that COVID-19 vaccines significantly protected adolescents from Delta's infections - without statistical significance (showed in table or figure) - this sentence means nothing !!!

The last sentence in the Conclusion is very important: Furthermore, VE should be continuously monitored to account for the possible rise of new VOCs as well as the development of new vaccines immunologically adapted to the predominant VOC. - but it must be confirmed in statistical calculations to be important for decision-makers, e.g., the Ministry of Health or the EMEA

Author Response

In my opinion, the keywords should be modified - add "age groups of children" instead “children” - attention to the negotiation!

We appreciate reviewer’s comment. We now modified the keywords.

“Policymakers in many countries” – (examples and references from other countries should be given, e.g., Germany, Poland, France) – “evaluated the risk-benefit ratios and progressively extended the COVID-19 vaccination campaign to underage people” – (it is worth citing other papers except 17-20 - do the papers concern only Italy?). 

We appreciate reviewer’s suggestion. We now added two sources: the first – from ECDC - states that as of 21 April 2022 all European countries have extended vaccination campaign to above 12 subjects, and 28 of them also recommend vaccination to 5-11 children; the second one – from HAS – is the official extension of vaccination to children 5-11 in France.

“In this context, the trade-off of vaccinating young people is less clear, and determining the effectiveness of COVID-19 vaccinations specifically on this subpopulation is crucial, to gain more insights on costs and benefits for evaluating and possibly reshaping public health policies”. – cite relevant items in pharmacoeconomics (e.g., cost-effectiveness, or EMA or HTA guidelines for public health policies). 

We appreciate reviewer’s suggestion. We now added a paper estimating cost-effectiveness of COVID-19 vaccination campaign.

“The descriptive characteristics of such study populations at the baseline were summarized through percentage and median with their respective interquartile range (IQR).” – it is also worth adding the value of means (the median does not always give a complete description of the data).

We agree with the reviewer’s suggestion. We now added mean values with their 95% C.I. for the number of swabs performed by each subject, which is the only continuous variables we had to deal with.

Materials and Methods

“Confidence intervals with a significance level equal to 5% (95% C.I.) were computed for each VE’s estimate. When an estimate turned out to be non-statistically significant, we indicated it just as non-significant (ns)”. – no information on the type and scope of the statistical tests used and the names of the software (except for the specific libraries of Python (Python 3.9.7). If only Python 3.9.7 is used, full justification should be given why. 

We apologize with the reviewer for the lack of clarity. We now tried to explain our approach throughout the material and methods section. Significance levels for vaccine effectiveness are directly estimated by the Cox regressions model, and they are by default based on the Wald test, which basically assess whether the unexponentiated coefficients are significantly different from 0. The significance immediately reflects on the confidence intervals of the estimates: when VE’s confidence interval cross the 0, the estimate turns out to be not significant, whereas when C.I. is entirely above or below 0 the estimate is statistically significant. Furthermore, the smaller the confidence interval the higher is the degree of significance of the corresponding estimate. We preferred to just present confidence interval estimates, because 1) in this way, we were able to keep light the results’ presentation and 2) for the above-mentioned reasons, Wald statistics did not provide any added qualitative information. This approach is the same followed f.e. by this remarkable study on vaccine effectiveness from Andrews et al.: Covid-19 Vaccine Effectiveness against the Omicron (B.1.1.529) Variant | NEJM. All the analysis – including Cox regression - was run in Python instead of SPSS (which is the other one statistical software we are used to utilize) because it guarantees very low execution time even with large amount of records. Figure 1 and Figure 2 were generated with Microsoft Power point and Microsoft Excel respectively. We now specify this information at the end of the material and methods section.

For the percentage results, you can use the structure tests - table 1 - a complete lack of statistical comparisons - they can be very interesting.

We apologize with the reviewer for the lack of clarity. We now will try to clarify this ambiguity. It must be noticed that the covariate on which we mostly focused on in the present study – vaccinal status – changes in time, whereas the purpose of Table 1 is just to summarize a snapshot of the descriptive characteristics of the study population as of the baseline by grouping subjects in those who will become positive at some point during the study period and those who will not.For instance, the 11th row of Table 1 should be read in this way: as of the baseline of the Delta period, 189,223 subjects were unvaccinated and 7,347 of them became positive before the end of the study period. Particularly, we could not infer that all these latter 7,347 were still unvaccinated when they become positive. Yet, looking at Figure 1 it turns out that just 6,913 of them tested positive before receiving their first dose. We brought to the light this example to explain that the lack of comparisons in Table 1 is intentional since in our opinion it would be methodologically incorrect and possibly misleading to compare percentage of positives looking at baseline characteristics who may change before the positivity itself.

Throughout the study, the levels of statistical significance should be given (sometimes even with no statistical difference) - then you can indicate the so-called statistical trends.

We apologize with the reviewer for the lack of clarity. As we now explained through the Material and Methods section, significance levels of VE estimates could be assessed by looking at the corresponding confidence interval – which directly reflect the result of Wald tests. The comparison between two different VE have been performed looking whether confidence intervals overlapped. Although naïf, this is a commonly used and very conservative (some groups could be significantly different even with overlapping, while without overlapping you can always infer significant difference) methods to infer statistically significant differences when just estimated value with their 95% confidence interval are available. We now added the conservativity of this approach a limitation of our study.

Figure 1 - no information about the individual colors' meaning (red, yellow, blue, navy blue).

We apologize with the reviewer for the lack of such information. Yet, there are no meanings for the choice of the colours in Figure 1. Indeed, they were chosen just to enlighten the category for which VE is presented in Figure 2. Furthermore, colours in Figure 1 coincide with corresponding categories’ colours in Figure 2.

Figures 1 and 2 - no information about the statistical significance or no statistical significance of it (the graph suggests that such significance exists). 

We apologize with the reviewer for the lack of clarity. As already mentioned, Cox regression model provided VE estimates with corresponding confidence interval, and significant estimates were compared looking at the overlapping of their corresponding confidence intervals. For instance, from Figure 2 one could conclude that the overall vaccine effectiveness during the Delta period is significantly higher than all the estimated effectiveness against Omicron, whereas during Omicron there is no statistical significance of differences in protection levels afforded by 2 doses among 5-11 and 3 doses among 12-17.

“We did not find clear-cut differences in protection levels” - “clear-cut differences” - question: are there any statistical differences? You have to write: clear-cut differences or not clear-cut differences or no statistical difference? Statistics is a very clear science.

We agree with the reviewer that we lack clarity in this point. We hope that at the light of our previous replies our statement would be now clearer. The fact is that looking at the VE estimates of a primary cycle with Pfizer and Moderna, it turns out that under the same timing from the last dose there are no significant differences in the afforded protection. As an example, it turns out that during the first week from vaccination, protection levels afforded by Pfizer and Moderna are respectively 81 (76 to 85) and 88 (81 to 92): thus, confidence intervals overlaps and no significantly difference may be deduced.

“We conducted statistically significant and robust analysis on large volumes of regional data.” - please indicate exactly which data is a statistically significant difference.

We hope that at the light of our previous replies our statement would be now clearer. All VE estimates for which confidence intervals have been reported – which represent most of the estimates – are statistically significant, and this is what we meant with our sentence.

Conclusion: we showed that COVID-19 vaccines significantly protected adolescents from Delta's infections - without statistical significance (showed in table or figure) - this sentence means nothing !!!

We now added our criteria for statistical significance, based on confidence intervals as a consequence of Wald test. The overall effectiveness during the Delta period was 94% (88 to 97), turning out to be highly significant.

The last sentence in the Conclusion is very important: Furthermore, VE should be continuously monitored to account for the possible rise of new VOCs as well as the development of new vaccines immunologically adapted to the predominant VOC. - but it must be confirmed in statistical calculations to be important for decision-makers, e.g., the Ministry of Health or the EMEA

We totally agree with the reviewer. We hope that with our modifications it is now clear the way we used to assess significance.

Reviewer 2 Report

 The authors used regional databases to retrospectively estimate vaccine effectiveness among children and adolescents. They found that COVID-19 vaccines were highly effective. The protection levels lasted longer and more protective  during a period of Delta variant predominance, rather than Omicron variant predominance. The presentation is good. The results are significant, which can provide scientific clue for controlling  COVID-19.

I have the following minor concerns.

(1) The methods seems too simple. It is just based on descriptive analysis. It could be better if there is more mechanism analysis.

(2) Is it possible to estimate how many cases had been prevented due to vaccination?

(3) What are the assumptions of the results?

Author Response

(1) The methods seems too simple. It is just based on descriptive analysis. It could be better if there is more mechanism analysis.

(2) Is it possible to estimate how many cases had been prevented due to vaccination?

(3) What are the assumptions of the results?

1) We appreciate reviewer’s comments. The purpose of our study was to investigate vaccine effectiveness among children during two different periods, adjusting estimates for available confounding factors. To achieve this goal, we adapted a simple but effective and commonly used approach, estimating the hazard ratio of vaccinal status through a regression model borrowed from survival analysis. Also, we stratified vaccine effectiveness per vaccinal status and timing from the last dose, making it possible to compare effectiveness of various vaccines and to investigate the waning effect in time. Even if we did not put in place advanced modelling techniques, we think that our study provides several added values than a simple descriptive analysis.

2) The estimation of the number of cases prevented due to vaccination is out of the purpose of our study, since we believe that reliable estimates may be obtained only with complex mathematical models. For instance, one may argue the number of prevented cases just by projecting the incidence of cases among unvaccinated to the whole population for the entire periods, but this approach would not consider too many factors (indirect effect on the epidemic’s spreading, heterogeneous hesitance to being tested, underascertainment…) to be reliable.

3) The main assumption for the presented results mainly depends by the use of Cox regression model. The goodness of the performance of this latter model is higher the nearer to the truth is the following assumption: the risk of the outcome (=infection) is proportional and homogeneous in time. Proportionality of the risk is clearly assumed, but we tried to get closer to this assumption by choosing period characterized by the same circulating VOC and always stratifying per the number of days elapsed from vaccination. The homogeneity of risk in time has been artificially provided by selecting two time periods for which the graph of the cumulative number of cases is as similar as possible to a straight line, meaning that each day counts approximately the same number of detected cases and that – consequently – the risk of being infected is as close as possible to be homogeneous in the whole time period.

Round 2

Reviewer 1 Report

Thank you very much for the corrected version of the paper. 

I see that most of my suggestions were implemented in the last version of the article. 

But: 

  1. There is no place (in scientific paper) for the so-called cases - if there is a relationship between the colors used in figs 1 and 2, they should be described under the figures. If it cannot, the colors should be removed. Second proposition: the text from the authors to me to be inserted under the figures or in the Limitations (of the paper). 

“We apologize with the reviewer for the lack of such information. Yet, there are no meanings for the choice of the colours in Figure 1. Indeed, they were chosen just to enlighten the category for which VE is presented in Figure 2. Furthermore, colours in Figure 1 coincide with corresponding categories’ colours in Figure 2.”

A significant part of the explanations received by the reviewer should be included in the Limitations (of the paper)(explanations of the lack of changes in the text of the publication)

This data should be abbreviated to Limitations (of the paper) - but must be included (short version):

„It must be noticed that the covariate on which we mostly focused on in the present study – vaccinal status – changes in time, whereas the purpose of Table 1 is just to summarize a snapshot of the descriptive characteristics of the study population as of the baseline by grouping subjects in those who will become positive at some point during the study period and those who will not. For instance, the 11th row of Table 1 should be read in this way: as of the baseline of the Delta period, 189,223 subjects were unvaccinated and 7,347 of them became positive before the end of the study period. Particularly, we could not infer that all these latter 7,347 were still unvaccinated when they become positive. Yet, looking at Figure 1 it turns out that just 6,913 of them tested positive before receiving their first dose. We brought to the light this example to explain that the lack of comparisons in Table 1 is intentional since in our opinion it would be methodologically incorrect and possibly misleading to compare percentage of positives looking at baseline characteristics who may change before the positivity itself.

Author Response

1) "We now added explanations about the colours in Figure 1 and their relation with those in Figure 2 between the two figures" and    2) "We summarized these informations in the limitations"